# Can We Solve 3D Vision Tasks Starting from A 2D Vision Transformer?

## Abstract

Vision Transformers (ViTs) have proven to be effective, in solving 2D image understanding tasks by training over large-scale image datasets; and meanwhile as a somehow separate track, in modeling the 3D visual world too such as voxels or point clouds. However, with the growing hope that transformers can become the "universal" modeling tool for heterogeneous data, ViTs for 2D and 3D tasks have so far adopted vastly different architecture designs that are hardly transferable. That invites an (over-)ambitious question: *can we close the gap between the 2D and 3D ViT architectures?* As a piloting study, this paper demonstrates the appealing promise to understand the 3D visual world, using a standard 2D ViT architecture, with only minimal customization at the input and output levels without redesigning the pipeline. To build a 3D ViT from its 2D sibling, we "inflate" the patch embedding and token sequence, accompanied with new positional encoding mechanisms designed to match the 3D data geometry. The resultant "minimalist" 3D ViT, named **Simple3D-Former**, performs surprisingly robustly on popular 3D tasks such as object classification, point cloud segmentation and indoor scene detection, compared to highly customized 3D-specific designs. It can hence act as a strong baseline for new 3D ViTs. Moreover, we note that pursuing a unified 2D-3D ViT design has practical relevance besides just scientific curiosity. Specifically, we demonstrate that Simple3D-Former naturally is able to exploit the wealth of pre-trained weights from large-scale realistic 2D images (e.g., ImageNet), which can be plugged into enhancing the 3D task performance "for free".

## 1 Introduction

In the past year, we have witnessed how transformers extend their reasoning ability from Natural Language Processing(NLP) tasks to computer vision (CV) tasks. Various vision transformers (ViTs) (Carion et al., 2020; Dosovitskiy et al., 2020; Liu et al., 2021b; Wang et al., 2022) have prevailed in different image/video processing pipelines and outperform conventional Convolutional Neural Networks (CNNs). One major reason that accounts for the success of ViTs is the self-attention mechanism that allows for global token reasoning (Vaswani et al., 2017b). It receives tokenized, sequential data and learns to attend between every token pair. These pseudo-linear blocks offer flexibility and global feature aggregation at every element, whereas the receptive field of CNNs at a single location is confined by small size convolution kernels. This is one of the appealing reasons that encourages researchers to develop more versatile ViTs, while keeping its core of self-attention module simple yet efficient, e.g., Zhou et al. (2021); He et al. (2021).

Motivated by ViT success in the 2D image/video space, researchers are expecting the same effectiveness of transformers applied into the 3D world, and many innovated architectures have been proposed, e.g., Point Transformer (PT, Zhao et al. (2021)), Point-Voxel Transformer (PVT, Zhang et al. (2021)), Voxel Transformer (VoTr, Mao et al. (2021)), M3DETR(Guan et al., 2021). Although most of the newly proposed 3D Transformers have promising results in 3D classification, segmentation and detection, they hinge on heavy customization beyond a standard transformer architecture, by either introducing pyramid style design in transformer blocks, or making heavy manipulation of self-attention modules to compensate for sparsely-scattered data. Consequently, ViTs for same type of vision tasks under 2D and 3D data is difficult to share similar architecture designs. On the other hand, there are recently emerged works, including Perceiver

IO(Jaegle et al., 2021a), and SRT(Sajjadi et al., 2022), that make fairly direct use of ViTs architecture, with only the input and output modalities requiring different pre-encoders.

That invites the question: *are those task-specific, complicated designs necessary for ViTs to succeed in 3D vision tasks? Or can we stick an authentic transformer architecture with minimum modifications, as is the case in 2D ViTs?* Note that the questions are of both scientific interest, and practical relevance. On one hand, accomplishing 3D vision tasks with standard transformers would set another important milestone for a transformer to become the universal model, whose success could save tedious task-specific model design. On the other hand, bridging 2D and 3D vision tasks with a unified model implies convenient means to borrow each other's strength. For example, 2D domain has a much larger scale of real-world images with annotations, while acquiring the same in the 3D domain is much harder or more expensive. Hence, a unified transformer could help leverage the wealth of 2D pre-trained models, which are supposed to learn more discriminative ability over real-world data, to enhance the 3D learning which often suffers from either limited data or synthetic-real domain gap. Other potential appeals include integrating 2D and 3D data into unified multi-modal/multi-task learning using one transformer (Akbari et al., 2021).

As an inspiring initial attempt, 3DETR (Misra et al., 2021) has been proposed. Despite its simplicity, it is surprisingly powerful to yield good end-to-end detection performance over 3D dense point clouds. The success of 3DETR implies that the reasoning ability of a basic transformer, fed with scattered point cloud data in 3D space, is still valid even without additional structural design. However, its 2D siblings, DETR(Devlin et al., 2019), cannot be naively generalized to fit in 3D data scenario. Hence, 3DETR is close to a universal design of but without testing itself over other 3D tasks, and embrace 2D domain. Moreover, concurrent works justify ViT can be extended onto 2D detection tasks without Feature Pyramid Network design as its 2D CNN siblings (Chen et al., 2021; Fang et al., 2022; Li et al., 2022), leading a positive sign of transferring ViT into different tasks. Uniform transformer model has been tested over multimodal data, especially in combination with 1D and 2D data, and some 3D image data as well (Jaegle et al., 2021b; Girdhar et al., 2022).

Therefore, we are motivated to design an easily customized transformer by taking a **minimalist** step from what we have in 2D, i.e., the standard 2D ViT (Dosovitskiy et al., 2020). The 2D ViT learns patch semantic correlation mostly under pure stacking of transformer blocks, and is well-trained over large scale of real-world images with annotations. However, there are two practical gaps when bringing 2D ViT to 3D space. i) **Data Modality Gaps**. Compared with 2D grid data, the data generated in 3D space contains richer semantic and geometric meanings, and the abundant information is recorded mostly in a spatially-scattered point cloud data format. Even for voxel data, the additional dimension brings the extra semantic information known as "depth". ii) **Task Knowledge Gaps**. It is unclear whether or not a 3D visual understanding task can gain from 2D semantic information, especially considering many 3D tasks are to infer the stereo structures(Yao et al., 2020) which 2D images do not seem to directly offer.

To minimize the aforementioned gaps, we provide a candidate solution, named as *Simple3D-Former*, to generate 3D understanding starting with a unified framework adapted from 2D ViTs. We propose an easy-to-go model relying on the standard ViT backbone where we made no change to the basic pipeline nor the self-attention module. Rather, we claim that properly modifying (i) positional embeddings; (ii) tokenized scheme; (iii) down-streaming task heads, suffices to settle a high-performance vision transformer for 3D tasks, that can also cross the "wall of dimensionality" to effectively utilize knowledge learned by 2D ViTs, such as in the form of pre-trained weights.

**Our Highlighted Contributions**

- We propose Simple-3DFormer, which closely follows the standard 2D ViT backbone with only minimal modifications at the input and output levels. Based on the data modality and the end task, we slightly edit only the tokenizer, position embedding and head of Simple3D-Former, making it sufficiently versatile, easy to deploy with maximal reusability.

- We are the first to lend 2D ViT's knowledge to 3D ViT. We infuse the 2D ViT's pre-trained weight as a warm initialization, from which Simple-3DFormer can seamlessly adapt and continue training over 3D data. We prove the concept that 2D vision knowledge can help further 3D learning through a unified model.

- Due to a unified 2D-3D ViT design, our Simple3D-Former can naturally extend to some different 3D down-streaming tasks, with hassle-free changes. We empirically show that our model yield competitive results in 3D understanding tasks including 3D object classification, 3D part segmentation, 3D indoor scene segmentation and 3D indoor scene detection, with simpler and mode unified designs.

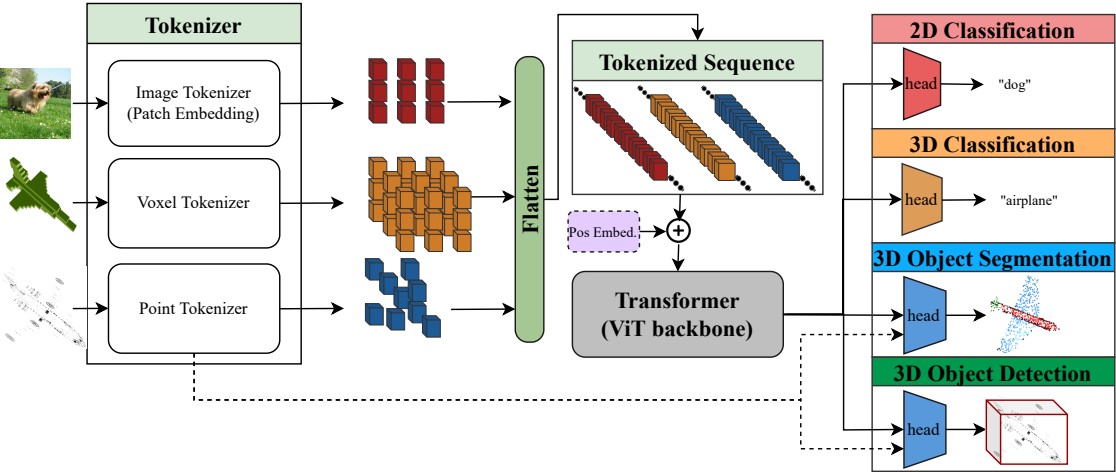

Figure 1: Overview of Simple3D-Former Architecture. As a Simple3D-Former, our network consists of three common components: tokenizer, transformer backbone (in our case we refer to 2D ViT), and a down-streaming task-dependent head layer. All data modalities, including 2D images, can follow the same processing scheme and share a universal transformer backbone. Therefore, we require minimal extension from the backbone and it is simple to replace any part of the network to perform multi-task 3D understanding. Dashed arrow refers a possible push forward features in the tokenizer when performing dense prediction tasks.

## 2 Related Work

### 2.1 Existing 2D Vision Transformer Designs

There is recently a growing interest in exploring the use of transformer architecture for vision tasks: works in image generation (Chen et al., 2020a; Parmar et al., 2018) and image classification (Chen et al., 2020a) learn the pixel distribution using transformer models. ViT (Dosovitskiy et al., 2020), DETR (Carion et al., 2020) formulated object detection using transformer as a set of prediction problem. SWIN (Liu et al., 2021b) is a more advanced, versatile transformer that infuses hierarchical, cyclic-shifted windows to assign more focus within local features while maintaining global reasoning benefited from transformer architectures. In parallel, the computation efficiency is discussed, since the pseudo-linear structure in a self-attention module relates sequence globally, leading to a fast increasing time complexity. DeIT (Touvron et al., 2021) focus on data-efficient training while DeepViT (Zhou et al., 2021) propose a deeper ViT model with feasible training. Recently, MSA (He et al., 2021) was introduced to apply a masked autoencoder to lift the scaling of training in 2D space. Recent works start exploring if a pure ViT backbone can be transferred as 2D object detection backbone with minimal modification, and the result indicates it might be sufficient to use single scale feature plus a Vanilla ViT without FPN structure to achieve a good detection performance (Chen et al., 2021; Fang et al., 2022; Li et al., 2022).

### 2.2 Exploration of 3D Vision Transformers

Transformer is under active development in the 3D Vision world (Fan et al., 2021; 2022). For example, 3D reconstruction for human body and hand is explored by the work (Lin et al., 2021) and 3D point cloud completion has been discussed in (Yu et al., 2021a). Earlier works such as Point Transformer (Engel et al., 2020) and Point Cloud Transformer (Guo et al., 2021) focus on point cloud classification and semantic

segmentation. They closely follow the prior wisdom in PointNet (Qi et al., 2017a) and PointNet++ (Qi et al., 2017b). These networks represent each 3D point as tokens using the Set Abstraction idea in PointNet and design a hierarchical transformer-like architecture for point cloud processing. Nevertheless, the computing power increases quadratically with respect to the number of points, leading to memory scalability bottleneck. Latest works seek an efficient representation of token sequences. For instance, a concurrent work PatchFormer (Cheng et al., 2021) explores the local voxel embedding as the tokens that feed in transformer layers. Inspired by sparse CNN in object detection, VoTR (Mao et al., 2021) modifies the transformer to fit sparse voxel input via heavy hand-crafted changes such as the sparse voxel module and the submanifold voxel module. The advent of 3DETR (Misra et al., 2021) takes an inspiring step towards returning to the standard transformer architecture and avoiding heavy customization. It attains good performance in object detection. Nevertheless, the detection task requires sampling query and bounding box prediction. The semantics contains more information from local queries compared with other general vision tasks in interest, and 3DETR contains transform decoder designs whereas ViT contains transformer encoder only. At current stage, our work focuses more on a simple, universal ViT design, i.e., transformer encoder-based design.

### 2.3 Transferring Knowledge between 2D and 3D

Transfer learning has always been a hot topic since the advent of deep learning architectures, and hereby we focus our discussion on the transfer of the architecture or weight between 2D and 3D models. 3D Multi View Fusion (Su et al., 2015; Kundu et al., 2020) has been viewed as one connection from Images to 3D Shape domain. A 2D to 3D inflation solution of CNN has been discussed in Image2Point (Xu et al., 2021), where the copy of convolutional kernels in inflated dimension can help 3D voxel/point cloud understanding and requires less labeled training data in target 3D task. On a related note, for video as a 2D+1D data, TimeSFormer (Bertasius et al., 2021) proposes an inflated design from 2D transformers, plus memorizing information across frames using another transformer along the additional time dimension. Liu et al. (2021a) provides a pixel-to-point knowledge distillation by contrastive learning.

It is also possible to apply a uniform transformer backbone in different data modalities, including 2D and 3D images, which is successfully shown by Perceiver (Jaegle et al., 2021b), Perceiver IO (Jaegle et al., 2021a), Omnivore (Girdhar et al., 2022), SVT (Sajjadi et al., 2022), UViM (Kolesnikov et al., 2022) and Transformer-M (Luo et al., 2022). All these works aim at projecting different types of data into latent token embedding but incorporate knowledge from different modalities either with self-attention or with cross-attention modules (with possibly one branch embedding from knowledge-abundant domain). Note that among all these aforementioned work. Only Perceiver discuss the application in point cloud modality with very preliminary result, and Omnivore discuss RGB-D data which is a primary version resembling 3D Voxel data. Contrary to prior works, our Simple3D-Former specifically aims at a model unifying 3D modalities, where point cloud and voxels are two most common data types that has not been extensively discussed in previous universal transformer model design. We discuss in particular how to design 3D data token embeddings as well as how to add 2D prior knowledge. In this paper, we show that with the help of a 2D vanilla transformer, we do not need to specifically design or apply any 2D-3D transfer step - the unified architecture itself acts as the natural bridge.

## 3 Our Simple3D-Former Design

### 3.1 Network Architecture

We briefly review the ViT and explain how our network differs from 2D ViTs when dealing with different 3D data modalities. We look for both voxel input and point cloud input. Then we describe how we adapt the 2D reasoning from pretrained weights of 2D ViTs. The overall architecture refers to Figure 1.

#### 3.1.1 Preliminary

For a conventional 2D ViT (Dosovitskiy et al., 2020), the input image $I \in \mathbb{R}^{H \times W \times C}$ is assumed to be divided into patches of size $P$ by $P$, denoted as $I_{x,y}$ with subscripts $x, y$. We assume $H$ and $W$ can be divided by $P$, thus leading to a sequence of length total length $N := \frac{HW}{P^2}$. We apply a *patch embedding* layers

$E : \mathbb{R}^{P \times P} \to \mathbb{R}^D$ as the tokenizer that maps an image patch into a $D$-dimensional feature embedding vector. Then, we collect those embeddings and prepend class tokens, denoted as $\boldsymbol{x}_{class}$, as the target classification feature vector. To incorporate positional information for each patch when flattened from 2D grid layout to 1D sequential layout, we add a positional embedding matrix $E_{pos} \in \mathbb{R}^{D \times (N+1)}$ as a learn-able parameter with respect to locations of patches. Then we apply $L$ transformer blocks and output the class labeling $\boldsymbol{y}$ by a head layer. The overall formula of a 2D ViT is:

$$\boldsymbol{z}_0 = [\boldsymbol{x}_{class}; E(I_{1,1}); \cdots ; E(I_{\frac{H}{P}, \frac{W}{P}})] + E_{pos}; \tag{1}$$

$$\tilde{\boldsymbol{z}}_l = MSA(LN(\boldsymbol{z}_{l-1})) + \boldsymbol{z}_{l-1}; \quad \boldsymbol{z}_l = MLP(LN(\tilde{\boldsymbol{z}}_l)) + \tilde{\boldsymbol{z}}_l; \tag{2}$$

$$\boldsymbol{y} = h(LN(\boldsymbol{z}_{L,0})). \tag{3}$$

Here $MSA$ and $MLP$ refer to the multi-head self-attention layer and multi-layer perception, respectively. The MSA is a standard qkv dot-product attention scheme with multi-heads settings (Vaswani et al., 2017a). The MLP contains two layers with a GELU non-linearity. Before every block, Layernorm (LN, Wang et al. (2019a); Baevski & Auli (2019)) is applied. The last layer class token output $\boldsymbol{z}_{L,0}$ will be fed into head layer $h$ to obtain final class labelings. In 2D ViT setting, $h$ is a single-layer MLP that maps $D$-dimensional class tokens into class dimensions (1000 for ImageNet).

The primary design principle of our Simple3D-Former is to keep transformer encoder blocks equation 2 same as in 2D ViT, while maintaining the tokenizing pipeline, equation 1 and the task-dependent head, equation 3. We state how to design our Simple3D-Former specifically with minimum extension for different data modalities.

### 3.1.2 Simple3D-Former of Voxel Input

We first consider the "patchable" data, voxels. We start from the data $V \in \mathbb{R}^{H \times W \times Z \times C}$ as the voxel of height $H$, width $W$, depth $Z$ and channel number $C$. We denote our 3D space tessellation unit as cubes $V_{x,y,z} \in \mathbb{R}^{T \times T \times T \times C}$, where $x, y, z$ are three dimensional indices. We assume the cell is of size $T$ by $T$ by $T$ and $H, W, C$ are divided by $T$. Let $N = \frac{HWZ}{T^3}$ be the number of total cubes obtained. To reduce the gap from 2D ViT to derived 3D ViT, we provide three different realizations of our Simple3D-Former, only by manipulating tokenization that has been formed in equation 1. We apply a same *voxel embedding* $E_V : \mathbb{R}^{T \times T \times T} \to \mathbb{R}^D$ for all following schemes. We refer readers to Figure 2 for a visual interpretation of three different schemes.

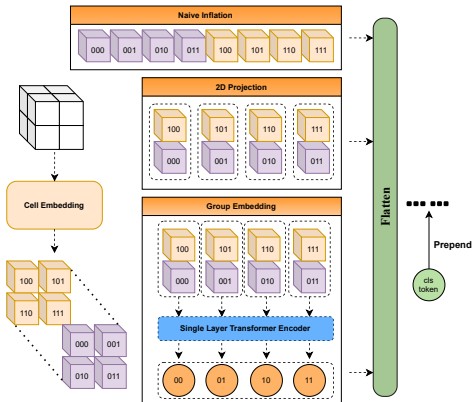

Figure 2: Three different voxel tokenizer designs. The given example is a $2^3$ cell division. We number cells for understanding. Top: Naive Inflation; We pass the entire voxel sequence in XYZ coordinate ordering. Middle: 2D Projection; We average along $Z$-dimension to generate 2D "patch" sequence unified with 2D ViT design. Bottom: Group Embedding; We introduce an additional, single layer transformer encoder to encode along $Z$-dimension, to generate 2D "group" tokens. Then the flattened tokenized sequence can thereby pass to a universal backbone.

**Naive Inflation** One can consider straight-forward inflation by only changing patch embedding to a voxel embedding $E_V$, and reallocating a new positional encoding matrix $E_{pos,V} \in \mathbb{R}^{(1+N) \cdot D}$ to arrive at a new tokenized sequence:

$$\boldsymbol{z}_0^V = [\boldsymbol{x}_{class}; E_V(\boldsymbol{x}_{1,1,1}); E_V(\boldsymbol{x}_{1,1,2}); \cdots ; E_V(\boldsymbol{x}_{1,2,1}); \cdots ; E_V(\boldsymbol{x}_{\frac{H}{T}, \frac{W}{P}, \frac{Z}{P}})] + E_{pos,V}. \tag{4}$$

We then feed the voxel tokenized sequence $\boldsymbol{z}_0^V$ to the transformer block equation 2. The head layer $h$ is replaced by a linear MLP with the output of probability vector in Shape Classification task.

**2D Projection (Averaging)** It is unclear from equation 4 that feeding 3D tokizened cube features is compatible with 2D ViT setting. A modification is to force our Simple3D-Former to think as if the data were

in 2D case, with its 3rd dimensional data being compressed into one token, not consecutive tokens. This resembles the occupancy of data at a certain viewpoint if compressed in 2D, and naturally a 2D ViT would fit the 3D voxel modality. We average all tokenized cubes if they come from the same XY coordinates (i.e. view directions). Therefore, we modify the input tokenized sequence as follows:

$$z_0^V = [\boldsymbol{x}_{class}; \frac{T}{Z}\sum_{z=1}^{\frac{Z}{T}} E(\boldsymbol{x}_{1,1,z}); \frac{T}{Z}\sum_{z=1}^{\frac{Z}{T}} E(\boldsymbol{x}_{1,2,z}); \cdots ; \frac{T}{Z}\sum_{z=1}^{\frac{Z}{T}} E(\boldsymbol{x}_{\frac{H}{T},\frac{W}{T},z})] + E_{pos,V}. \tag{5}$$

The average setting consider class tokens as a projection in 2D space, with $E_{pos,V} \in \mathbb{R}^{(1+\tilde{N})\cdot D}, \tilde{N} = \frac{HW}{T^2}$, and henceforth $E_{pos,V}$ attempts to serve as the 2D projected positional encoding with $\tilde{N}$ "patches" encoded.

**Group Embedding**  A more advanced way of tokenizing the cube data is to consider interpreting the additional dimension as a "word group". The idea comes from group word embedding in BERT-training (Devlin et al., 2019). A similar idea was explored in TimesFormer (Bertasius et al., 2021) for space-time dataset as well when considering inflation from image to video (with a temporal 2D+1D inflation). To train an additional "word group" embedding, we introduce an additional 1D Transformer Encoder(TE) to translate the inflated Z-dim data into a single, semantic token. Denote $\boldsymbol{V}_{x,y,-} = [\boldsymbol{V}_{x,y,1}; \boldsymbol{V}_{x,y,2}; \cdots ; \boldsymbol{V}_{x,y,\frac{Z}{P}}]$ as the stacking cube sequence along $z$-dimension, we have:

$$\tilde{E}(\boldsymbol{V}_{x,y,-}) := TE(E(\boldsymbol{V}_{x,y,-})), \tag{6}$$

$$z_0^V = [\boldsymbol{x}_{class}; \tilde{E}(\boldsymbol{V}_{1,1,-}); \cdots ; \tilde{E}(\boldsymbol{V}_{\frac{H}{P},\frac{W}{P},-})] + \mathbf{E}_{pos,V}. \tag{7}$$

Here $\tilde{E}$ as a compositional mapping of patch embedding and the 1D Transformer Encoder Layer (TE). The grouping, as an "projection" from 3D space to 2D space, maintains more semantic meaning compared with 2D Projection.

### 3.1.3  Simple3D-Former of Point Cloud Data

It is not obvious how one can trust 2D ViT backbone's reasoning power applied over point clouds, especially when the target task changes from image classification to dense point cloud labeling. We show that, in our Simple3D-Former, a universal framework is a valid option for 3D semantic segmentation, with point cloud tokenization scheme combined with our universal transformer backbone. We modify the embedding layer $E$, positional encoding $E_{pos}$ and task-specific head $h$ originated from equation 1 and equation 3 jointly. We state each module's design specifically, but our structure does welcome different combinations.

**Point Cloud Embedding**  We assume the input now has a form of $(X, P), X \in \mathbb{R}^{N\times3}, P \in \mathbb{R}^{N\times C}$, referred as point coordinate and input point features. For a given point cloud, we first adopt a MLP (two linear layers with one ReLU nonlinearity) to aggregate positional information into point features and use another MLP embedding to lift point cloud feature vectors. Then, we adopt the same Transition Down (TD) scheme proposed in Point Transformer (Zhao et al., 2021). A TD layer contains a set abstraction downsampling scheme, originated from PointNet++ (Qi et al., 2017b), a local graph convolution with kNN connectivity and a local max-pooling layer. We do not adopt a simpler embedding only (for instance, a single MLP) for two reasons. i) We need to lift input point features to the appropriate dimension to transformer blocks, by looking loosely in local region; ii) We need to reduce the cardinality of dense sets for efficient reasoning. We denote each layer of Transition Down operation as $TD(X, P)$, whose output is a new pair of point coordinate and features $(X', P')$ with fewer cardinality in $X'$ and lifted feature dimension in $P'$. To match a uniform setting, we add a class token to the tokenized sequence. Later on, this token will not contribute to the segmentation task.

**Positional Embedding for Point Clouds**  We distinguish our positional embedding scheme from any previous work in 3D space. The formula is a simple addition and we state it in equation 8. We adopt only a single MLP to lift up point cloud coordinate $X$, and then we sum the result with the point features $P$ altogether for tokenizing. We did not require the transformer backbone to adopt any positional embedding components to fit the point cloud modality.

**Segmentation Task Head Design** For dense semantic segmentation, since we have proposed applying a down-sampling layer, i.e. TD, to the input point cloud, we need to interpolate back to the same input dimension. We adopt the Transition Up(TU) layer in Point Transformer (Zhao et al., 2021) to match TD layers earlier in tokenized scheme. TU layer receives both input coordinate-feature pair from the previous layer as well as the coordinate-feature pair from the same depth TD layer. Overall, the changes we made can be formulated as:

$$\tilde{P} = MLP_2(P + MLP_1(X)); \quad \boldsymbol{z}_0^{PC} = [\boldsymbol{x}_{class}; TD(TD(X, \tilde{P}))]; \tag{8}$$

$$\boldsymbol{y} = h(TU(TU(LN(\boldsymbol{z}_{L,1:N}), TD(X, \tilde{P})), (X, \tilde{P}))). \tag{9}$$

We refer to Figure 3 as the overall visualized design of our Simple3D-Former for point cloud data. For detail architecture of Simple3D-Former in segmentation, we refer readers to Appendix C.

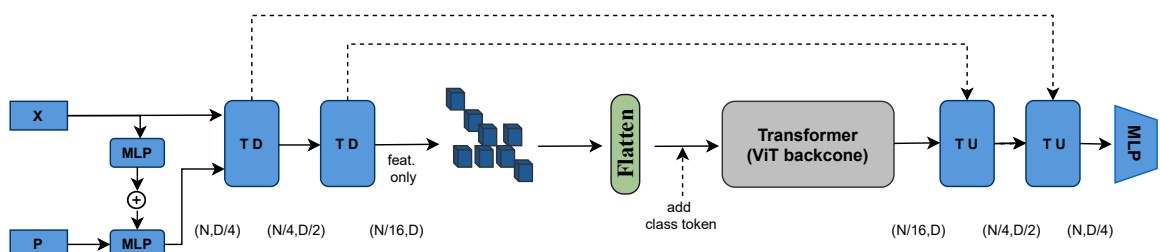

Figure 3: To transfer from a classification backbone into an object part segmentation backbone, we propose some additional, yet easy extensions that fit into 3D data modality. Given input point clouds with its coordinates $X$, features $P$, we compose positional information into features first and use a simple MLP to elevate features into $D/4$ dimensions, given $D$ the dimension of backbone. Then we apply two layers of Transition down over pair $(X, P)$, then feed the abstracted point cloud tokens sequentially into the transformer backbone. To generate the dense prediction. We follow the residual setting and add feature output from TD layers together with the previous layers' output into a transition up layer. Then we apply a final MLP layer to generate dense object part predictions.

## 3.2 Incorporating 2D Reasoning Knowledge

The advantage of keeping the backbone transformer unchanged is to utilize the comprehensive learnt model in 2D ViT. Our Simple3D-Former can learn from 2D pretrained tasks thanks to the flexibility of choice of backbone structure, without any additional design within transformer blocks. We treat 2D knowledge as either an initial step of finetuning Simple3D-Former or prior knowledge transferred from a distinguished task. The overall idea is demonstrated in Figure 4.

**Pretraining from 2D ViT** As shown in Figure 1, we did not change the architecture of transformer backbone. Therefore, one can load transformer backbone weight from 2D-pretrained checkpoints with any difficulty. This is different from a direct 3D Convolutional Kernel Inflation (Shan et al., 2018; Xu et al., 2021) by maintaining the pure reasoning from patch understanding. We observed that one needs to use a small learning rates in first few epochs as a warm-up fine-tuning, to prevent catastrophic forgetting from 2D pretrained ViT. The observation motivates a better transfer learning scheme by infusing the knowledge batch-by-batch.

**Retrospecting From 2D Cases by Generalization** As we are transferring a model trained on 2D ImageNet to unseen 3D data, retaining the ImageNet domain knowledge is potentially beneficial to the generalized 3D task. Following such a motivation, we require our Simple3D-Former to memorize the representation learned from ImageNet while training on 3D. Therefore, apart from the loss function given in 3d task $\mathcal{L}_{3d}$, we propose adding the divergence measurement as a proxy guidance during our transfer learning process (Chen et al., 2020b). We fix a pretrained teacher network (teacher ViT in Figure 4). When training

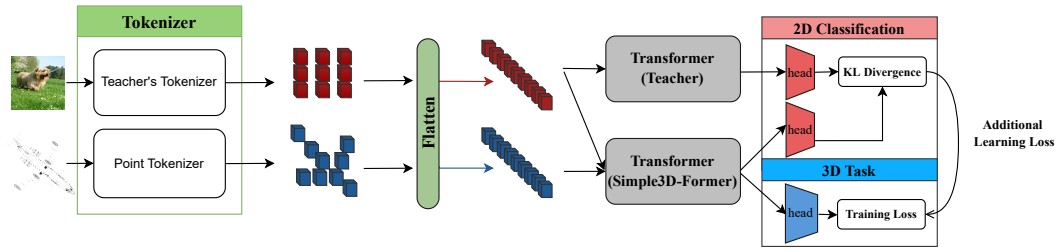

Figure 4: Memorizing 2D knowledge. The teacher network (with all weights fixed) guide the current task by comparing the performance over the pretrained task.

Table 1: Baseline Comparison in 3D Object Classification

| Method | Modality | ModelNet40 | | ScanObjectNN |
| --- | --- | --- | --- | --- |
| | | mAcc.(%) | OA. (%) | PB-T50-RS OA. (%) |
| VoxelNet(Maturana & Scherer, 2015) | Voxel | 83.0 | 85.9 | - |
| PointNet(Qi et al., 2017a) | Point | 86.2 | 89.2 | 68.0 |
| PointNet++(Qi et al., 2017b) | Point | - | 91.9 | 77.9 |
| Perceiver(Jaegle et al., 2021b) | Point | - | 85.7 | - |
| DGCNN (Wang et al., 2019b) | Point | 90.2 | 92.2 | 78.1 |
| Image2Point(Xu et al., 2021) | Voxel | - | 89.1 | - |
| Point Transformer(Zhao et al., 2021) | Point | 90.6 | 93.7 | 81.2 |
| PVT(Zhang et al., 2021) | Point | - | 94.0 | - |
| Point-BERT(Yu et al., 2021b) | Point[1] | 93.2 | - | 83.1 |
| **Simple3D-Former** (ours)[2] | Voxel(NI) | 82.8 | 86.5 | - |
| | Voxel(Avg.) | 82.4 | 85.9 | - |
| | Voxel(GE) | 84.0 | 88.0 | - |
| | Point | 89.3 | 92.0 | 83.1 |

[1] We report the result with 1024 point sample inputs here to match with other methods.

[2] NI:Naive Embedding; Avg.: Averaging; GE: Group Embedding.

each mini-batch of 3D data, we additionally bring a mini-batch of images from ImageNet validation set (in batch size $M$). To generate a valid output class vector, we borrow every part except Transformer blocks from 2D teacher ViT and generate 2D class labeling. We then apply an additional KL divergence to measure knowledge memorizing power, denoted as:

$$\mathcal{L} := \mathcal{L}_{3d} + \lambda \sum_{i=1}^{M} KL(\boldsymbol{y}_{teacher} || \boldsymbol{y}). \tag{10}$$

The original 3d task loss, $\mathcal{L}_{3d}$ with additional KL divergence regularization, forms our teacher-student's training loss. The vector $\boldsymbol{y}_{teacher}$ is from 2D teacher ViT output, and $\boldsymbol{y}$ comes from a same structure as teacher ViT, with the transformer block weight updated as we learn 3D data. In practical implementation, since the teacher ViT is fixed, the hyper-parameter $\lambda$ depends on the task: see Section 4.

## 4 Experiments

We test our Simple-3DFormer over three different 3D tasks: object classification, semantic segmentation and object detection. For detailed dataset setup and training implementations, we refer readers to Appendix A.

### 4.1 3D Object Classification

3D object classification tasks receives a 3D point cloud or 3D voxel as its input and output the object categories. We test 3D classification performance over ModelNet40 (Wu et al., 2015) dataset and ScanObjectNN (Uy et al., 2019) dataset. To generate voxel input of ModelNet40, We use binvoxMin (2004 - 2019); Nooruddin & Turk (2003) to voxelize the data into a $30^3$ input. The size 30 follows the standard setup in ModelNet40 setup. We choose to apply Group Embedding scheme as our best Simple3D-Former to compare with existing state-of-the-art methods. We further report the result in Table 1, compared with other state-of-the-art

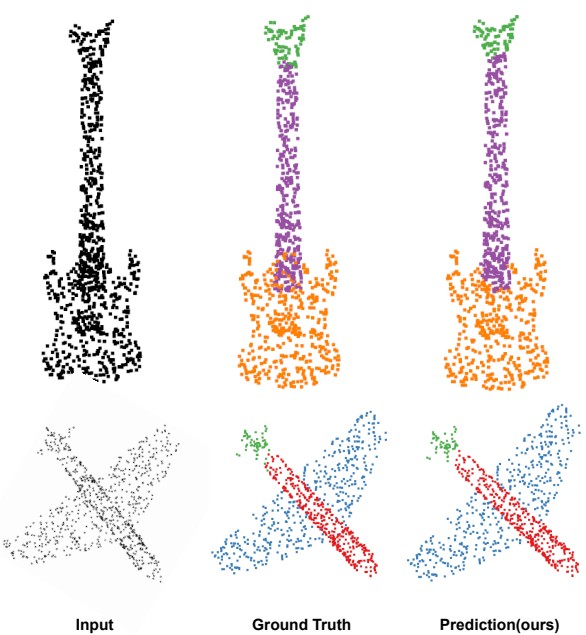

**Input**          **Ground Truth**          **Prediction(ours)**

Figure 5: Selective visualizations of point cloud part segmentation.

Table 2: Comparison of 3D segmentation results on the ShapeNetPart and S3DIS dataset.

| Method | ShapeNetPartSeg | | S3DIS | |
|---|---|---|---|---|
| | cat. mIoU.(%) | ins. mIoU.(%) | mAcc.(%) | ins. mIoU.(%) |
| PointNet(Qi et al., 2017a) | 80.4 | 83.7 | 49.0 | 41.1 |
| PointNet++(Qi et al., 2017b) | 81.9 | 85.1 | - | - |
| PointCNN(Li et al., 2018) | 84.6 | 86.1 | 75.6 | 65.4 |
| DGCNN(Wang et al., 2019b) | 82.3 | 85.1 | 56.1 | - |
| KPConv(Thomas et al., 2019) | 85.1 | 86.4 | 72.8 | 67.1 |
| Point Transformer(Zhao et al., 2021) | 83.7 | 86.6 | 76.5 | 70.4 |
| PVT(Zhang et al., 2021) | - | 86.5 | 67.7 | 61.3 |
| PatchFormer(Cheng et al., 2021) | - | 86.7 | - | 68.1 |
| **Simple3D-Former** (ours) | 83.3 | 86.0 | 72.5 | 67.0 |

methods over ModelNet40 dataset, and over ScanObjectNN dataset. We optimize the performance of our Simple3D-Former with voxel input by setting up $T = 6$ in equation 7. We finetune with pretrained weight as well as using memorizing regularization equation 10 with $M$ equal to batch size. The classification result of point cloud modality is generated by dropping out two TU layers and passing the class token into a linear classifier head, with the same training setup as ShapeNetV2 case. Our network outperforms previous CNN based designs, and yields a competitive performance compared with 3D transformers. Several prior works observed that adding relative positional encoding within self-attention is important for a performance boost. We appreciate these findings, but claim that a well-pretrained 2D ViT backbone, with real semantic knowledge infused, does assist a simple, unified network to learn across different data modalities. The observation is particularly true over ScanObjectNN dataset, where transformer-enlightened networks outperform all past CNN based networks. Our method, with relatively small parameter space, achieves a similar result compared with Point-BERT.

## 4.2 3D Point Cloud Segmentation

3D point cloud segmentation is a two-fold task. One receives a point cloud input (either an object or indoor scene scans) and output a class labels per input point within the point cloud. The output simultaneously contains segmentation as well as classification information. Figure 5 is a visual example of object part segmentation task. We report our performance over object part segmentation task in Table 2. The target datasets

Table 3: 3D Detection Result over SUN RGB-D data

| Metric | BoxNet | VoteNet | 3DETR | 3DETR-masked | H3DNet | **Simple3D-Former** (ours) |
|---|---|---|---|---|---|---|
| $AP_{25}$ | 52.4 | 58.3 | 58.0 | 59.1 | 60.1 | 57.6 |
| $AP_{50}$ | 25.1 | 33.4 | 30.3 | 32.7 | 39.0 | 32.0 |

are ShapeNetPart (Yi et al., 2016) dataset and Semantic 3D Indoor Scene dataset, S3DIS (Armeni et al., 2016). We do observe that some articulated desiged transformer network, such as Point Trasnformers (Zhao et al., 2021) and PatchFormer (Cheng et al., 2021) reach the overall best performance by designing their transformer networks to fit 3D data with more geometric priors, while our model bond geometric information only by a positional embedding at tokenization. Nevertheless, our model does not harm the performance and is very flexible in designing. Figure 5 visualizes our Simple3D-Former prediction. The prediction is close to ground truth and it is surprisingly coming from 2D vision transformer backbone without any further geometric-aware infused knowledge. Moreover, the prior knowledge comes only from ImageNet classification task, indicating a good generalization ability within our network.

### 4.3 3D Object Detection

3D object detection is a pose estimation task. For any given 3D input, one needs to return a 3D bounding box of each detected objects of targeted class. In our experiments we use point cloud input data. We test our simple-3DFormer for SUN RGB-D detection task (Song et al., 2015). We compare our results with BoxNet (Qi et al., 2019), VoteNet (Qi et al., 2019), 3DETR (Misra et al., 2021) and H3DNet (Yin et al., 2020). We follow the experiment setup from Misra et al. (2021): we report the detection performance on the validation set using mean Average Precision (mAP) at IoU thresholds of 0.25 and 0.5, referred to as AP25 and AP50. The result is shown in Table 3 and the evaluation is conducted over the 10 most frequent categories for SUN RGB-D. Even though 3DETR is a simple coupled Transformer Encoder-Decoder coupled system, we have shown that our scheme can achieve similar performance by replacing 3D backbone with our simple3D-Former scheme.

### 4.4 Ablation Study

**Different Performance With/Without 2D Infused Knowledge** To justify our Simple3D-Former can learn to generalize from 2D task to 3D task, we study the necessity of prior knowledge for performance boost. We compare the performance among four different settings: i) train without any 2D knowledge; ii) with pretrained 2D ViT weights loaded; iii) with a teacher ViT only, by applying additional 2D tasks and use the loss in equation 10; iv) using both pretraining weights and a teacher ViT.

Results shown in Table 4 reflect our motivation. One does achieve the best performance by not only infusing prior 2D pretraining weight at an early stage, but also getting pay-offs by learning without forgetting prior knowledge. It probably benefits a more complex, large-scale task which is based upon our Simple3D-Former.

Table 4: Power of 2D Prior Knowledge, with 2D projection scheme and evaluated in OA. (%) The performance is tested under ShapeNetV2 and ModelNet40 dataset.

| Pretrain Usage | ShapeNetV2 | ModelNet40 |
|---|---|---|
| Without Any 2D Knowledge | 82.8 | 86.5 |
| With 2D pretraining | 83.5 | 86.6 |
| Teacher ViT | 84.3 | 87.6 |
| Pretrain + Teacher ViT | **84.5** | **88.0** |

Table 5: Power of 2D Prior Knowledge (with teacher ViT) in 3D task, evaluated in cat. mIOU.(%) and ins. mIoU.(%) over ShapeNet Part Segmentation

| 3D Data Portion | M | cat. mIoU.(%) | ins, mIoU. (%) |
|---|---|---|---|
| 25% | 0 | 79.1 | 83.3 |
| | 32 | 79.4 | 83.1 |
| | 64 | 79.8 | 83.6 |
| 50% | 0 | 79.5 | 84.1 |
| | 32 | 79.9 | 84.0 |
| | 64 | 80.3 | 84.5 |
| 100% | 0 | 81.1 | 84.6 |
| | 32 | 82.8 | 85.4 |
| | 64 | 83.1 | 85.7 |

**Performance in Low-Quantity 3D Data Regime**  The computation complexity of point cloud data goes up as the number of sample points blows up. Even for a fixed point cloud sampling of 1024 points, it is inefficient to train over the entire dataset. To test the generalization ability of our Simple3D-Former, we perform a test to explore the power of 2D knowledge transferring. We use only a portion of training data in 3D and change the batch size of source task images $M$ at different scales. Result in Table 5 justify the performance over point cloud part segmentation task. Though one needs more data to attain higher accuracy, we found 2D pretrained knowledge offers an accuracy boost. The result indicates a potential joint-training across different data modality and different tasks to find universal transformers with good generalization ability.

### 4.5   Limitation of our work

Our method challenges the necessity of heavy-lifting design of transformers for 3D tasks. However, a potential drawback of our simple3D-Former is an overlook in 3D-aware only knowledge in the tokenizing process. It has been justified in Point Transformer(Zhao et al., 2021) the layer-wise positional encoding is beneficial for point cloud understanding. While our method is flexible in choosing tokenizer and transformer backbone, the performance might be hindered from a strict fixed transformer structure.

Another concern is that the performance of our method is strongly related with the complexity of selected transformer backbone. Our Simple3D-Former outperforms early-stage point cloud CNN architectures with a total of 29.59G Multiply Add Cumulations(MACs). On the contrary, Point Transformer Zhao et al. (2021) has a total of 36.76G MACs. The detailed model complexity comparison is shown in Appendix B and D. We observe that even with the most complex ViT model (deit-base) we have been testified, we cannot yield the state-of-the-art performance compared with concurrent works. It is in particular true for large indoor scene data (S3DIS) shown in Table 2. How to yield the best trade-off and how the result is different from choice of backbone (especially the embedding dimension) need to be analized further by introduing different designs of transformer backbones.

## 5   Conclusion

We retrospect the development of Vision Transformer and propose a unified version of a 3D transformer, named as Simple3D-Former, that learns from 2D rich-knowledge domain. a 2D ViT can inflate into a 3D ViT, by replacing 2D feature embedding, positional embedding and end-task head layer. Moreover, we justify that 2D domain knowledge helps our model perform better when understanding 3D data and the power of our model can be further strengthened by learning without forgetting. Our experimental result indicates self-attention modules, if learnt from 2D domain knowledge, can be distilled and thereby help the learning 3D object classification, part segmentation and detection tasks under both voxel and point cloud data. In the subsequent work, we hope to explore more versatile combinations of 2D transformer backbones attached with distinguished 3D feature extracting layers, and include complex tasks in large scale datasets.

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

# A    Dataset Setup and Implementation Details

## A.1    3D Object Classcification

**Dataset Setup**    ModelNet40 consists of 12311 samples with 9843 training samples and 2468 test samples. It contains 40 classes in total. The original data is aligned and in point-cloud format. ScanObjectNN contains 2902 CAD objects with background knowledge provided in point cloud as well. It contains 15 classes in total. We apply our model over the augmented PB_T50_RS batch samples, in which bounding boxes of objects can shift up to 50% and objects are perturbed with rotation and scaling, resulting in 14510 total input train/test

samples. We follow the standard sampling scheme to generate a subset of 1024 points for every point cloud model in both datasets. We additionally use ShapeNetV2[47] to testify our Simple3D-Former for voxel input as well. For details on ShapeNetV2 classification, we refer readers to Appendix B.

**Implementation Details**   We use one TITAN A100 for training. For voxel classification task, we use the Adam optimizer with an initial warm-up at starting learning rate of 0.01, which is decayed by a factor of 0.5 every 20 epochs. The batch size is set to 64. We trained 100 epochs in total. The hyperparameter $\lambda$ is set to 0.1 back in (10). We evaluate mean of class-wise accuracy (mAcc), and overall point-wise accuracy (OA). The voxel embedding $E_V$ we choose is a single convolutional layer with kernel size of $T$ and stride $T$ to generate tokenized sequence and remain simple. The pretrained knowledge comes from DeIT. The experiment is conducted with DeIT-base backbone with ImageNet-1K pretraining of image size 224. We justify the ablation study for choosing backbones and appropriate positional embedding parameters for optimal performance. In addition, we show that optimal performance is obtained by using not only ViT backbone but pretrained 2D knowledge and the help of memorizing 2D tasks. We report the result in Appendix B accordingly.

### A.2   3D Point Cloud Segmentation

**Dataset Setup**   For object part segmentation, we test over ShapeNetPart dataset, containing $16,881$ pre-aligned shapes with dense labeling of 50 different parts over 16 distinguished categorical objects. We sample 1024 points for every point cloud model with standard process. We evaluate our Simple3D-Former over semantic indoor scene semantic segmentation dataset, S3DIS, as well. S3DIS contains 5 large-scale indoor scans with 12 semantic elements. We use area 5 as the test case while the reamining areas are treated as training data. We sample 4096 points for every partitioned indoor scene with standard process.

**Implementation Details**   The training is conducted on one TITAN A100. For object part segmentation task, we use the SGD optimizer with an initial learning rate of 0.05, which is decayed by a factor of 0.1 every 100 epochs. The batch size is set to 64. We trained our Simple3D-Former up-to 300 epochs. For semantic segmentation task , we use the SGD optimizer with an initial learning rate of 0.1, which is decayed by a factor of 0.5 every 20 epochs. We trained 100 epochs with batch size 8. We use DeIT-base as the backbone ViT for both tasks. The hyperparameter $\lambda$ is set to 0.1 back in equation 10. We evaluate categorical mean intersection over union (cat. mIOU.) and instance mean intersection over union (ins. mIOU.) respectively for ShapeNetPart dataset while we report the mean accuracy and instance mean intersection over union in S3DIS dataset.

### A.3   3D Point Cloud Object Detection

**Dataset Setup**   We apply our Simple3D-Former onto a standard 3D indoor detection benchmark, SUN RGB-D (v1). SUN RGB-D contains 5000 training samples with oriented bounding box annotations while KITTI dataset contains 7518 raw 3d input.

**Implementation Details**   To justify our simple3D-Former can be embedded naturally into a detection model's 3D backbone, we modify 3DETR's backbone into our version, while keep the decoder head unchanged. The training is conduced on one TITAN A100 and trained over 1080 epochs. Detailed architecture of Simple3D-Former backbone used in detection task is explained in Appendix C.

## B   More Classification Result With Voxel Input

ShapeNetV2 dataset contains 52456 samples from 55 categories and we use a fixed $80\% - 20\%$ train-test split throughout our experiments. The voxel data is of size $128^3$. Note that ShapeNetV2 is evaluated only when we determine which Simple3D-Former setup optimizes the performance over voxel data.

It has been explored in 2D ViT the relationship between the size of patches and the classification accuracy over image dataset. There is no such prior belief in 3D voxel data, so we test our Simple3D-Formers under different settings to find the optimal scheme. For point cloud input, we fix our model all from the beginning.

Table 6: Performance of Different Simple-3DFormer Design on ShapeNetV2 Classification evaluated on OA. (%), either with pretrained 2D ViT weight guidance (W P.) or without pretrained 2D ViT guidance (W/O P.).

| Scheme | Token Length | Naive Transformer | |
|---|---|---|---|
| | | W P. | W/O P. |
| Naive Inflation | 8 by 8 by 8 | 83.1 | 79.8 |
| 2D Projection | 8 by 8 | 83.6 | 82.3 |
| Group Embedding | 8 by 8 | 85.0 | 84.9 |
| Naive Inflation | 14 by 14 by 14 | 85.5 | 85.5 |
| 2D Projection | 14 by 14 | 83.5 | 82.8 |
| Group Embedding | 14 by 14 | 87.6 | 86.8 |

Table 6 shows the preliminary result. We test under two different cell size settings: $T = 16$ (8 cells per axis) and $T = 9^1$ (14 cells per axis) in equation 4, equation 5 and equation 7 . Among all configurations, Group Embedding outperforms Naive Inflation and 2D Projection. More importantly, the pretraining weight adopted in transformer backbone before training over 3D data does help to improve the accuracy of object classification. Another observation is that the size of token sequence affects the result as well. A $9^3$ cell yields more semantic meaning compare to that of a $16^3$ cell, which neglects too many local connections. Hence, in the following experiments over voxel data,

We further justify that among all transformer backbone mimic from 2D ViT siblings, DeIT-base attains optimal performance. The result is shown in Table 7.

Table 7: Different 2D ViT backbone performance and complexity comparison. The table shows our Simple3D-Former under Group Embedding setup.

| Backbone Name | ImageNet(2D) | ModelNet40(3D) | | |
|---|---|---|---|---|
| | Param. (M) | FLOPs (G) | Param. (M) | OA. (%) |
| DeiT-tiny | 5 | 0.28 | 5 | 84.5 |
| DeiT-small | 22 | 1.12 | 21 | 86.7 |
| DeiT-base | 86 | 4.46 | 85 | 88.0 |

**Different Performance Under Particular Ordering** When discussing 2D Projection tokenized scheme for voxel data, we implicitly assume we project along $Z$-dim. We show that we are not biased from the choice of ordering. Table 8 explains different result of particular ordering in 2D Projection scheme, in both ShapeNetV2 and ModelNet40 dataset. We denote the ordering $XYZ$ as the normal input order, where $Z$-dim data is projected or grouped. Similarly, $YZX$ refers to the $X$-dim data projection and $ZXY$ refers to the $Y$-dim data projection. The result indicates the optimal choice of projection is dataset dependent, but the performance is optimal further when considering group embedding scheme.

Table 8: Performance under different ordering of input voxels, with 2D Projection scheme and evaluated in OA. (%)

| Projection View | ShapeNetV2 | ModelNet40 |
|---|---|---|
| XYZ | 83.6 | 82.1 |
| YZX | 84.5 | 83.2 |
| ZXY | 81.9 | 84.3 |

## C   Detailed architecture of Simple3D-Former In Point Cloud Modality

**Simple3D-Former for Part Segmentation Task** The overall Simple3D-Former of point cloud segmentation has a different design of data tokenizer and downstream head (to produce information not from class tokens). In point tokenizer part, two layers of point set abstractions are applied. The Transition Down (TD) layer comes from Point Transformer. A TD layer contains a set abstraction downsampling scheme, originated from PointNet++, a local graph convolution with kNN connectivity, and a local max-pooling layer.

---

[1]For $T = 9$ Group Embedding scheme, we use batch size of 32 instead.

Rather than adding relative positional embedding in attention layers as most 3d-aware transformers did, we propose to add the relative positional embedding in local convolution layers in pointnet++ skeleton (i.e. PointSetAbstraction operation in PointNet++), to avoid artificial design in transformer attention modules, but incorporate local embeddings beforehand. Each TD layer reduces the number of points by 4 with a 2$x$ scale-up of the embedding dimension. The newly distilled point tokenized sequence is then fed into the ViT backbone. The Transition Up (TU) layer comes from Point Transformer as well. It interpolates over the original point coordinates by neighboring features and scales down the embedding dimension by 2. TU module also contains a residual block that adds the point feature vectors back in the corresponding TD layer, resulting in a U-Net architecture. We provide code snippets in Listing 1 and 2 for readers to match the practical implementation with Figure 3.

**Simple3D-Former for 3D Detection Task**  In our 3D detection experiment, we replace 3DETR's transformer encoder structure into our Simple3D-Former design, and generate the output head with same structure as in 3detr, and fix other part to show the flexibility of our design.

```
self.transition_downs = nn.ModuleList()
for i in range(2):
    channel = self.embed_dim // 4 * 2 ** (i+1)
    self.transition_downs.append(TransitionDown(npoints // 4 ** i, nneighbor, [channel // 2
    + 3, channel, channel]))

self.transition_ups = nn.ModuleList()
for i in reversed(range(2)):
    channel = self.embed_dim // 4 * 2 ** i
    self.transition_ups.append(TransitionUp(channel * 2, channel, channel))

self.fc1 = nn.Sequential(
    nn.Linear(d_points, self.embed_dim // 4),
    nn.ReLU(),
    nn.Linear(self.embed_dim // 4, self.embed_dim // 4)
)

self.fc_pos_embed = nn.Sequential(
    nn.Linear(3, self.embed_dim // 4),
    nn.ReLU(),
    nn.Linear(self.embed_dim // 4, self.embed_dim // 4)
)
```

Listing 1: Code Snippet to define TD/TU layers and two MLPs

```
def forward_features(self, x):
    xyz, f= x[...,:3], self.fc1(x)
    f = self.pos_drop(f + self.fc_pos_embed(xyz))

    xyz_0, points_0 = ...
        self.transition_downs[0](xyz, f)
    xyz_1, points_1 = ...
        self.transition_downs[1](xyz_0, points_0)
    x = points_1

    # Add dummy class tokens to mimic ViT's style
    cls_token = self.cls_token.expand(x.shape[0], -1, -1)
    x = torch.cat((cls_token, x), dim=1)

    for blk in self.blocks:
        x = blk(x)
    x = self.norm(x)
    x = x[:, 1:]
    x = self.transition_ups[0](xyz_1, x, xyz_0, points_0)
    x = self.transition_ups[1](xyz_0, x, xyz, f)
    return x.mean(1)
```

Listing 2: Code Snippet in forward function

# D    Different Point Cloud Simple3D-Former Design

We additionally show different results regarding the number of TD/TU coupled layers as the ablation study of Simple3D-Former structure. Note that if TD/TU layer number is 0, only a MLP layer is applied to lift input point cloud features and another MLP is applied to encode absolute positions. Moreover, we fix two MLP layers back in Eqn. (10) to have the same output dimension for a reasonable comparison, when testing with 0 or 1 layer TD/TU setting. For 2 layer TD/TU setup, the dimension of MLP is changing according to the embedding dimension $D/4$ based on different choices of backbones: DeIT-tiny, $D = 192$; DeIT-small, $D = 384$; DeIT-base, $D = 768$.

As TD layer scales up the embedding dimension of point vectors while reducing the size of tokenized sequence, different ViT backbones, when equipped with same number of TD/TU layers, have different scalings. The experiment setup is the same as described in Section 4.2, with $M = 64$ for 2D knowledge infusing. All setups applied pretrained weight from the corresponding backbones as well.

Table 9 shows the result regarding different TD/TU layers. We found that by introducing point abstraction, the performance of a 2D pretrained ViT backbone can be further improved compared with MLP only setup (0 TD/TU layers), with the total computational cost relatively lower. This reflects the claim back in Section 3, where we point out the necessity of point cloud data modality modification to fit into the universal transformer backbone. Our Simple3D-Former can adapt from the change of data modality and obtain a good result, compare with CNN-based schemes.

Table 9: Different Simple3D-Formers' performance over part segmentation task, evaluated in cat. mIoU. (%), ins. mIoU. (%) and MACs. (G).

| # of TD/TU Layers | ShapeNetPart | | |
|---|---|---|---|
| | cat. mIoU. (%) | ins. mIoU. (%) | MACs. (G) |
| 0 (DeIT-tiny) | 81.7 | 84.7 | 5.53 |
| 1 (DeIT-small) | 82.9 | 85.1 | 6.53 |
| 1 (DeIT-base) | 82.5 | 84.9 | 26.04 |
| 2 (DeIT-tiny) | 82.2 | 84.7 | 1.87 |
| 2 (DeIT-small) | 82.5 | 84.7 | 7.42 |
| 2 (DeIT-base) | 83.1 | 85.7 | 29.59 |

