# OpenReview forum: "Can We Solve 3D Vision Tasks Starting from A 2D Vision Transformer?"
_TMLR — Rejected by TMLR_

### Review · Reviewer_8Ndb · 2023-01-02

**Summary Of Contributions:**

The paper proposed a novel transformer architecture for 3D vision tasks that leverages pre-trained 2D ViT features. This paper is one of the first papers that attempts to borrow fully-trained 2D features to 3D vision tasks. Experimental results demonstrated that the proposed approach performs reasonably well, and ablation study shows that the pre-trained 2D features were truly effective.

**Audience:**

Yes

**Broader Impact Concerns:**

The applications of this paper is standard computer vision tasks (classification, segmentation, detection). Any broader impact concerns that are applicable to those can be applied to this paper.

**Claims And Evidence:**

Yes

**Requested Changes:**

I'm currently leaning towards a rejection because of the bigger concerns I was having. However it would be interesting to at least see further analysis on the experimental results comparing with concurrent approaches.

**Strengths And Weaknesses:**

[Strengths]
- The paper addresses an important and challenging task of 3D vision application using pre-trained 2D vision transformer features. The proposed algorithm is a step towards this ambitious goal.
- Experiments of the paper are decent, there are ablation studies to demonstrate the effectiveness of each individual blocks of designs.
- The writings of this paper is actually pretty nice. I was able to follow the description of the work well, and the diagrams of the paper is proper.

[Weaknesses]
- Despite the great ambition of this work, the experimental results didn't really shine. This is OK, however I see little analysis on why the performance didn't match or surpass existing approaches (failure cases, visualizations, etc). Without those data, it's not fair to claim that this is truly the very first "baseline" algorithm for this task.
- However, my biggest concern is whether the settings of this paper is truly the next step for incorporating 2D VIT features to 3D -- the difference of "pretrained 2D ViT on imageNet" and "3D applications on ShapeNet" is big: there are 1) dataset statistics differences, 2) data type (2D vs 3D) differences, and subsequently the application differences. The paper attempts to address 1+2 together with Simple3D-Former, but I fear the stride is too big. I almost feel like the paper could consider dealing with RGB-D data (or at least 3D data with corresponding images that loosely related to the input), and figure out a way to incorporate RGB ViT features to 3D data+tasks.
- Another concern of mine is the 3D data representations -- given 3D voxels/point cloud, without proper orientations and such, I don't really know 1) whether the output is prone to input orientations using simple3D-Former, and 2) if so, how to actually use the algorithm in real applications with unknown 3D poses.

---

> ### Author Response · Authors · 2023-01-23
> **Response to Reviewer 8Ndb**
>
> We thank Reviewer 8Ndb for the comment and suggestions for our paper. We list out our response to each point that the reviewer has been addressed.
>
> ### Why the performance didn't match or surpass existing approaches.
>
> For existing approaches that incoporating transformer models to solve versatile tasks involving 3D data, they are prone to do heavy-lifting design to fit the modality.  Rather, transformer structure shall be a natural bridge across modality without too much changes, like CLIP Model has achieved. These motivates us to propose Simple3D-Former, not a complex 3D transformer.
>
> Our goal **is not to beat** the SOTA performance in 3D vision tasks (although it would be impressive to achieve that using a very simple model). Rather, our goal is to show i) 3D Task with 3D input, if tokenized, can be fit into a self-attention module (i.e. transformers) which is originally applied in 2D data; ii) With the help of 2D pretraining and 2D teacher task(which is relatively easy), one can use the pretrained transformers as backbones to apply to 3D tasks with minimum extension. The performance might be improved if one apply different tokenization or transformer backbone.
>
> ### The difference of "pretrained 2D ViT on imageNet" and "3D applications on ShapeNet" is big. I almost feel like the paper could consider dealing with RGB-D data (or at least 3D data with corresponding images that loosely related to the input), and figure out a way to incorporate RGB ViT features to 3D data+tasks.
>
> We appreciate this question and agree that this is a promising direction for a potential improvement. [1] and [2] are concurrent works that are working on this particular direction and shows the potential of closing the gap between "pretrained 2D ViT on imageNet" and "3D applications on ShapeNet". Our work, though simple, directly fits into the scope.
>
> ### The 3D data representations.
>
> > 1) whether the output is prone to input orientations using simple3D-Former
>
> In the main content of the paper we currently only apply group embedding in the canonical space (the normal orientation view). Table 4 discusses 2D projection scheme only and in the ShapeNet dataset, the +Y is the up orientation and -Z is the front orientation and therefore the X-dim projection is more powerful compared with other directions. We have been dedicating to a better trained model and we provide an additional ablation study to show that group embedding scheme is working on ShapeNet differently based on which direction you are viewing as the additional 1D information. We report our current model performance in the following table:
>
> | View Direction      | OA. (%) |
> | ----------- | ----------- |
> | XYZ (XY plane)      | 87.41       |
> | YZX (YZ plane)   | 87.65        |
> | ZXY (ZX plane)   | 87.32 |
>
>
> The result yields two conclusions: i) YZ plane contains the information, this is consistent with Table 4 in Appendix. ii) More importantly, consider 2D+1D group embedding can reduce the gap between different view directions, compared with naive projection where one only average the data along the third dimension. We will update the result in our revisions.
>
>
> > 2) if so, how to actually use the algorithm in real applications with unknown 3D poses.
>
> Pose detection itself is a challenging task. Fortunately, for point cloud modality, we train the network with data augmentation in scaling, rotation and transtition and therefore point cloud data is more robust with respect to input in arbitrary pose. For a real world data with unknown 3D poses, one has to actively learn it. Our scheme can be combined with previous works with upright orientation learning in both voxel[3] and point cloud[4] modalities.
>
> [1] Dong, Runpei, Zekun Qi, Linfeng Zhang, Junbo Zhang, Jianjian Sun, Zheng Ge, Li Yi, and Kaisheng Ma. "Autoencoders as Cross-Modal Teachers: Can Pretrained 2D Image Transformers Help 3D Representation Learning?." arXiv preprint arXiv:2212.08320 (2022).
>
> [2] Yao, Yuan, Yuanhan Zhang, Zhenfei Yin, Jiebo Luo, Wanli Ouyang, and Xiaoshui Huang. "3D Point Cloud Pre-training with Knowledge Distillation from 2D Images." arXiv preprint arXiv:2212.08974 (2022).
>
> [3] Zishun Liu, Juyong Zhang, Ligang Liu. "Upright orientation of 3D shapes with Convolutional Networks." Graph. Model. 85: 22-29 (2016).
>
> [4] Pang, Xufang, Feng Li, Ning Ding, and Xiaopin Zhong. "Upright-Net: Learning Upright Orientation for 3D Point Cloud." In Proceedings of the IEEE/CVF Conference on Computer Vision and Pattern Recognition, pp. 14911-14919. 2022.

---

### Review · Reviewer_8aTN · 2023-01-03

**Summary Of Contributions:**

In this paper, the authors investigate the feasibility of adapting a 2D vision transformer, a model typically used for image processing tasks, to handle 3D input data such as voxels and point clouds. For voxels, they propose a method for converting 3D data into sequences that can be processed by the 2D vision transformer, with only minimal modifications. The authors demonstrate the effectiveness of this approach and the benefits of using a pretrained 2D vision transformer for 3D data processing tasks.

**Audience:**

No

**Broader Impact Concerns:**

N.A.

**Claims And Evidence:**

No

**Requested Changes:**

The reported number of other methods needs to be carefully checked.

It would be helpful if the authors can clearly compares the difference between the proposed method and Point-Transformer.

Figure 2 is good. It can be make even more clear if the author can color different patches by different color.

**Strengths And Weaknesses:**

(+) The motivation is clear and relevant. It is important to make the broader community be aware of what a simple vision transformer can achieve in various tasks. This paper quantitatively studies this aspect.

(+) This paper proposes one candidate solution to use a pretrained 2D ViT structures for 3D tasks, and shows it’s better than naively finetuning from the model.

(-) Some of the experiment results are confusing. The performance of other methods seem to be inconsistent with the original paper. For example, Point-BERT reports >93.2 accuracy in their original paper (Table 1 of the Point-BERT paper) while this paper reports 84.1. The number 84.1 seems to be the part segmentation results (Table 3 of Point-BERT paper) instead. Besides, DGCNN is also a point-based method.

(-) The distinctions between “Simple3D-Former of Point Cloud Data” and point-transformer are not clear. They both use the Transition Down and Transition Up design. It seems the only difference is the way of position embedding. However, there is no ablation about that and the proposed method does not perform better than Point Transformer (ModelNet40, ShapeNetPartSeg, and S3DIS).

**Are the claims made in the submission supported by accurate, convincing and clear evidence?**

This paper explicitly states three major contributions: 1) the development of a 2D vision transformer that can process 3D data with minimal modifications; 2) a method for transferring knowledge from a 2D vision transformer to a 3D model using a teacher-student KL divergence loss; and 3) the ability of the proposed architecture to be easily adapted to downstream tasks such as point cloud detection and segmentation.

Claim 1 is well supported. This paper shows voxels can be easily embedded to a sequence of tokens and processed by a vanilla ViT.

Claim 2 may not be very convincing. Image2Point shows 2D pretrained model can help with 3D recognition. The new things in this paper is to do that using ViT and a teacher-student KL divergence loss. I believe this is a good verification of this idea but may not be a surprising finding to the community.

Claim 3 is not very clear. The down-streaming tasks are point-cloud detection and segmentation. However, in the point-cloud case, the whole network structure is similar to point-transformer which can also do segmentation without many changes.

In summary, I don’t think the claims are completely supported by the network design and empirical results.

**Would some individuals in TMLR's audience be interested in the findings of this paper?**

I think there are certain points that interest audiences of TMLR. However, the main information is overlapped with previous methods (Image2Point and Point Transformer). I don’t think the presented results are surprising and motivating.

---

> ### Author Response · Authors · 2023-01-23
> **Response to Reviewer 8aTN**
>
> We appreciate reviewer 8aTN for appreciating that our proposed method has **clear and relavant motivation**.
>
> > It is important to make the broader community be aware of what a simple vision transformer can achieve in various tasks. This paper quantitatively studies this aspect.
>
> For your major concerns, we have updated our documents and put further analysis within the main context of our manuscript and hereby
>
> ###  Some of the experiment results are confusing
>
> We do apologize that our reported number for point-BERT ModelNet40 classification result is incorrect. We have checked our reported number and correct the misleading number and DGCNN modality typo.
>
> ### Distinction between point-Transformer and our Simple3D-Former under point cloud modality
>
> We have explained our network architecture explicitly in Appendix C. Our Simple3D-Former is a tokenizer-transformer architecture where the tokenized scheme and what one could add as downstreaming head is detached from transformer layers. On the other hand, **point transformer is adding transformer layers after each transition down and transition up blocks**.  Our tokenizer is a composition of transition down layers, aiming to embed point cloud sequences into tokenized sequences. Besides, the positional embedding in point transformer was taken place layer-by-layer while our Simple3D-Former apply positional embedding only once when tokenzing the data. We have added Figure 5 shown in Appendix back to main content for better explaining our network structure.
>
> The major concern is that our experimental result may imply no advantage by doing a universal,flexible design other than a point-cloud transformer under a particular design for point cloud modality. Our method take point transformer's transition up and transition down blocks as an example of one point cloud encoder(tokenizer) and one point cloud downstream head(or decoder). It can be replaced by any structure, for instance, recently developed PointNeXT[1].
>
> [1] Qian, Guocheng, Yuchen Li, Houwen Peng, Jinjie Mai, Hasan Abed Al Kader Hammoud, Mohamed Elhoseiny, and Bernard Ghanem. "PointNeXt: Revisiting PointNet++ with Improved Training and Scaling Strategies." Advances in Neural Information Processing Systems (2022).
>
> ### Change of Figure 2
>
> Thanks for your suggestion! We have updated figure 2 for better visualization purpose of our scheme.

---

### Review · Reviewer_X4rG · 2023-01-09

**Summary Of Contributions:**

The paper builds on the premise that current transformer architectures for 3D tasks are commonly customized to the specific task, and in particular not directly compatible with pre-trained 2D Vision Transformers (ViT).
In order to reach this goal, the Simple3D-Former framework is proposed that contains a pre-trained 2D ViT at its core. Different input modalities (2D images, point clouds, voxel grids) can be fed into this 2D ViT model through a "Tokenizer" that encodes the input into a set of tokens and outputs are obtained through readout heads. The specific design of tokenizer and readout heads depends on the specific modality and task.
Instead of, or in addition to pretraining the ViT model, the authors also investigate a proxy guidance loss proposed by Chen et al., 2020b that is meant to retain 2D-image feature statistics while fine-tuning the model for the 3D task.

The model is evaluated on the 3D tasks of object classification, semantic segmentation and
object detection. Simple3D-Former, with appropriate adjustments for each task and data modality, appears to be competitive with prior works in these tasks.
Finally, ablation studies show that using 2D pretraining, employing the proxy loss, and the combination thereof each lead to small but noticeable improvements in metrics.


**Audience:**

Yes

**Broader Impact Concerns:**

I do not have specific broader impact concerns.

**Claims And Evidence:**

Yes

**Requested Changes:**

Main points

* The premise of the paper requires an overhaul to focus the scope of the paper to the subset of 3D vision that it tackles. This relates both to the set of tasks and the types of representations that Scene3D-Former can be easily applied to.

* Misleading claims such as being the first to apply 2D pre-training for 3D tasks should be modified. "ViTs for 2D and 3D tasks have so far adopted vastly different architecture designs that are hardly transferable" is too strong of a statement and can be challenged. Several works including the already mentioned 3DETR [Misra et al.], Perceiver IO [2], and SRT [3] make fairly direct use of ViTs architecture, with only the input and output modalities requiring different pre-encoders (akin to the "tokenizer") and decoders (readout heads). In particular, SRT directly applies the normal (hybrid) ViT model on patched 2D RGB conditioning. Further tasks that have an RGB input but 3D outputs are also often solved with fairly vanilla ViT architectures (e.g., depth estimation).

* The related works section should be expanded, and highly related works such as Perceiver / IO [1, 2] should be extensively discussed and compared against (in terms of approach).

* I strongly recommend the authors to include a discussion on limitations.


Smaller points

* Given that the improvements in Tab. 4 are fairly small, it would be good to know the expected variance for this experiment, especially for the "Without Any 2D Knowledge" baseline that may have the largest variance. Have the authors investigated this, and could error margins be reported?

* The authors may be interested in a recent work with a similar spirit of unifying model architectures, focusing on 2D tasks, but including depth estimation based on ViT [4].

* The experimental section would benefit from a brief description of the respective task in the accompanying section.

* "The transformer backbone can get multi-head self-attention pretrained weight loaded without any difficulties, exactly as any 2D ImageNet pretraining loading." — I assume *all* ViT model weights are loaded from the 2D-pretrained checkpoint, not just the self-attention weights? The authors may want to tweak this sentence to clarify.

* It would be interesting for the reader to report changes in convergence speed for the ablations in Table 4. Does pre-training only (slightly) improve final performance, or does it lead to significantly faster convergence as well?

* "We choose to apply Group Embedding scheme as our best Simple3D-Former to compare with existing state-of-the-art methods." Since the authors propose 3 variants, they should report results for all voxel tokenizer models.

* Suggestion: Figure 1 might be better placed 1-2 pages lower to be closer to the context.

* Page 5, 1st paragraph: it might be better to explicitly mention that "dot-product" attention is being used

* The manuscript would benefit from another proofreading pass. Some non-exhaustive notes:
  * pg 1, bottom: self-attentions -> self-attention
  * pg 2, 3rd paragraph: leading -> Leading
  * pg 2, 4th paragraph: when bring -> when bringing
  * pg 4, 2nd paragraph: Preceiver -> Perceiver
  * pg 5, 2nd paragraph: "specifically" is duplicated
  * pg 7, 2nd paragraph: Convectional -> Convolutional
  * pg 10, conclusion 2nd sentence: capitalization


References:
1. Jaegle et al. Perceiver: General Perception with Iterative Attention.
2. Jaegle et al. Perceiver IO: A General Architecture for Structured Inputs & Outputs.
3. Sajjadi et al. Scene Representation Transformer: Geometry-Free Novel View Synthesis Through Set-Latent Scene Representations.
4. Kolesnikov et al. UViM: A Unified Modeling Approach for Vision with Learned Guiding Codes.



**Strengths And Weaknesses:**

Strengths

* Unifying architectures between 2D and (different) 3D tasks follows recent trends and is a good idea, both to simplify future research and to allow large-scale pre-trained models to be employed across domains.

* It is good to see the model achieve similar results as the baselines. The authors do not overstate the results, and SOTA is indeed not a requirement for this work.

* The ablations are meaningful and interesting.


Weaknesses

* The premise of the paper is not very concrete, harming the delivery of the paper. 3D vision is a wide field that is much larger than the ubiquitous classification, segmentation and detection tasks, which are often solved with similar or the same model architecture, as is the case in this manuscript. Similarly, 3D representations find many more shapes than point clouds and voxels. Especially with the current popularity of Neural Rendering [1] and NeRF [2, 3], relating this work to implicit representations and how they could be used for 3D-aware downstream applications (in particular with Simple3D-Former) merits a discussion.

* The authors mention: "At current stage, our work focuses more on a simple, universal ViT design, i.e., transformer encoder-based design". This is of course fine, though it should be noted that the readout heads used in Simple3D-Former are a type of decoder (albeit not transformer-based). Here, it should be noted that pure encoder-only models could have shortcomings in efficiency or scalability, as certain tasks may require querying a large number of items at the decoder stage which can be much more efficient in an encoder-decoder setting.

References:
1. Tewari et al. Advances in Neural Rendering
2. Mildenhall et al. NeRF: Representing Scenes as Neural Radiance Fields for View Synthesis.
3. Gao et al. NeRF: Neural Radiance Field in 3D Vision, A Comprehensive Review.

---

> ### Author Response · Authors · 2023-01-23
> **Response to Reviewer X4rG - major requests**
>
> We appreciate reviewer X4rG for all valuable suggestions and think our work. We have addressed response of all points below. For updates of all bulletin points, please check our revisioned manuscript.
>
> ### The premise of the paper requires an overhaul to focus the scope of the paper to the subset of 3D vision that it tackles.
>
> We presume you use "Scene3D-Former" to refer our "Simple3D-Former" work. In our paper we provide results for tasks including 3D object classfications, 3D part segmentations, 3D indoor scene segmentations and 3D in door scene detections. Sometimes we might overclaim/imply other universal tasks can be achieved under Simple3D-Former without proper proof. We have changed the over claimed  tone in Introduction and conclusion to specifically discuss within the scope of what we have evaluated in the manuscript.
>
> One particular topic that reviewer mentioned is the recent success in neural rendering:
>
> > Especially with the current popularity of Neural Rendering [1] and NeRF [2, 3], relating this work to implicit representations and how they could be used for 3D-aware downstream applications (in particular with Simple3D-Former) merits a discussion.
>
> We believe the novel view synthesis task is different from all tasks that our Simple3D-Former can tackle. First, Neural radiance field is a per-instance optimization (its generalization ability is still underway), whereas what we originally realize in this paper are common 3D tasks that requires to solve a problem by probing large number of 3D datas and find commonality (e.g. class labels). Secondly, the particular task is a 2D multi-view images input to a generative model that can yield images from unseen view, given multi-view images as prior knowledge. There are several works that already discuss the usage of transformer, we list one here as an example:
>
> > Xu, Dejia, Yifan Jiang, Peihao Wang, Zhiwen Fan, Humphrey Shi, and Zhangyang Wang. "SinNeRF: Training Neural Radiance Fields on Complex Scenes from a Single Image." arXiv preprint arXiv:2204.00928 (2022).
>
> We wish to address our point that novel-view synthesis is a next-level topic involving per-scene optimization and generative prior. Henceforth, we shall not discuss it in our manuscript at the current stage. However, we thank the reviewer's suggestion of lowering our tone and pointing out this interesting direction to follow.
>
>
>
> ### The related works section should be expanded, and highly related works such as Perceiver / IO [1, 2] should be extensively discussed and compared against (in terms of approach).
>
> We have added a paragraph in Section 2.3 that discuss further how our method aims at briding 2D and 3D as well as design in particular 3D data token embeddings, which is different focus compared with previous works. We would like to add more detailed comparison with each modality each paper consider with a tabled version in our final manuscript.
>
>
> ### Misleading claims such as being the first to apply 2D pre-training for 3D tasks should be modified.
>
> Thanks for the correction. We have changed the misleadning sentence and add appropriate citations for those pioneer works.
>
>
> ### Include a discussion on limitations
>
> We add Section 4.5 right before the conclusion to expand and discuss limitation of our work.

---

> ### Author Response · Authors · 2023-01-23
> **Response to Reviewer X4rG - minor requests**
>
> ### Minor edition requests
>
> > Given that the improvements in Tab. 4 are fairly small, it would be good to know the expected variance for this experiment, especially for the "Without Any 2D Knowledge" baseline that may have the largest variance. Have the authors investigated this, and could error margins be reported?
>
> Due to limited computing resource, we only run these examples in three different random seeds and report them in the following table. We cannot tell if "Without Any 2D Knowledge" yields a significant larger variance at now. Nevertheless, we will incoporate a more detailed analysis in further revisions.
>
> | Pretrain Usage | ModelNet40_1 | ModelNet40_2 | ModelNet40_3 |
> | ----------- | ----------- | ---------- |  ---------- |
> |Without Any 2D Knowledge|86.5| 85.4 | 86.2|
> |With 2D pretraining|86.6| 86.0  | 86.7 |
> |Teacher ViT|87.6|  87.1| 86.9|
> |Pretrain + Teacher ViT|88.0 | 87.5 | 87.7
>
>
>
> > The authors may be interested in a recent work with a similar spirit of unifying model architectures, focusing on 2D tasks, but including depth estimation based on ViT [4].
>
> Appreciate the UViM paper. We have cited this work in our manuscript.
>
> > The experimental section would benefit from a brief description of the respective task in the accompanying section.
>
> We have added intro for each task in the accompanying section.
>
>
> > "The transformer backbone can get multi-head self-attention pretrained weight loaded without any difficulties, exactly as any 2D ImageNet pretraining loading." — I assume all ViT model weights are loaded from the 2D-pretrained checkpoint, not just the self-attention weights? The authors may want to tweak this sentence to clarify.
>
> Yes, what you claim is correct. We have rewritten this sentence for clarification.
>
> > Does pre-training only (slightly) improve final performance, or does it lead to significantly faster convergence as well?
>
> For our observation, there is no significant convergence speed up for starting from pre-trained in finding the best result. However, we do observe the change of first few epochs is drastically different, as pre-trained weight might flucuate the performance of first few epochs, pre-training with teacher guidance is helping to find the near-optimal most. We will add a detailed analysis in our final version of the manuscript.
>
> > Report results for all voxel tokenizer models
>
> We have updated Table 1 in Section 4 to reflect this change. Ablation study result using ShapeNetV2 data was reported in Table 6 in Appendix.
>
> > Suggestion: Figure 1 might be better placed 1-2 pages lower to be closer to the context.
>
> We have moved Figure 1 one page lower.
>
> > Page 5, 1st paragraph: it might be better to explicitly mention that "dot-product" attention is being used
>
> The clarifiation has already been added in the manuscript.
>
> > The manuscript would benefit from another proofreading pass
>
> Thanks for all these detailed proofreading effort. We have updated the manuscript after another pass of proofreading and highlight these changes.

---

### Decision · Action_Editors · 2023-03-20

**Recommendation:** Reject

**Comment:**

Three experts in the community review the paper. All three reviewers thank the authors' efforts in the rebuttal. However, two reviewers remain unconvinced about the claim of unifying different 3D tasks using a simple and well-established model. More specifically, Reviewer 8Ndb is concerned about the un-grounded claim and weak experimental result. Reviewer 8aTN is concerned about the difference between the proposed method and Image2Point in terms of the idea and point-transformer in terms of network structure. Reviewer X4rG appreciates the work and believe that using 2D-ViT models for 3D tasks is valuable for the community. But also discussed the applicability of the proposed implementation presented implementation.

Given these concerns shared among the reviewers, the AE does not find ground to accept the paper and believes that the paper is not yet ready for publication.

**Audience:**

Yes, there are audiences who will find the topic of the work interesting.

**Claims And Evidence:**

As discussed by the reviewers, there are three key claims in the paper.

1) 2D vision transformer can process 3D data.
- This is well supported as voxels can be converted to a sequence of tokens.

2) transferring knowledge from 2D vision transformer to 3D.
- Image2Point also shows how 2D model can help 3D recognition.

3) Downstream applications:
- The proposed network architecture is similar to point-transformer (which can also perform semantic segmentation).